# Sound generation in zebrafish with Bio-Opto-Acoustics

Itia A. Favre-Bulle [1,2✉], Michael A. Taylor [3], Emmanuel Marquez-Legorreta [2], Gilles Vanwalleghem [2], Rebecca E. Poulsen [2], Halina Rubinsztein-Dunlop [1] & Ethan K. Scott [2✉]

Hearing is a crucial sense in underwater environments for communication, hunting, attracting mates, and detecting predators. However, the tools currently used to study hearing are limited, as they cannot controllably stimulate specific parts of the auditory system. To date, the contributions of hearing organs have been identified through lesion experiments that inactivate an organ, making it difficult to gauge the specific stimuli to which each organ is sensitive, or the ways in which inputs from multiple organs are combined during perception. Here, we introduce Bio-Opto-Acoustic (BOA) stimulation, using optical forces to generate localized vibrations in vivo, and demonstrate stimulation of the auditory system of zebrafish larvae with precise control. We use a rapidly oscillated optical trap to generate vibrations in individual otolith organs that are perceived as sound, while adjacent otoliths are either left unstimulated or similarly stimulated with a second optical laser trap. The resulting brain-wide neural activity is characterized using fluorescent calcium indicators, thus linking each otolith organ to its individual neuronal network in a way that would be impossible using traditional sound delivery methods. The results reveal integration and cooperation of the utricular and saccular otoliths, which were previously described as having separate biological functions, during hearing.

[1] School of Mathematics and Physics, The University of Queensland, Brisbane, Australia. [2] Queensland Brain Institute, The University of Queensland, Brisbane, Australia. [3] Australian Institute for Bioengineering and Nanotechnology, The University of Queensland, Brisbane, Australia. ✉email: i.favrebulle@uq.edu.au; ethan.scott@uq.edu.au

Evolution has produced diverse approaches for hearing. Understanding different auditory systems in nature provides insights into the role of hearing in ecology, and has provided valuable design information for biomimetic microphone technologies[1,2]. While mammalian hearing is based on a single organ, the cochlea, which only senses pressure waves, animals such as fish[3], crustaceans[4], and insects[5] all have multiple sensory organs that collectively provide hearing.

In adult fish, sound is sensed through the combined contributions of the otoliths (or "ear stones"), the swim bladder, and lateral line[6–12]. Otoliths, which are attached to sensory hair cells, move and vibrate in response to mechanical waves, which allow the detection of acceleration and variable sound frequencies. The swim bladder, in addition to regulating buoyancy, expands and compresses as pressure waves pass, and in "hearing specialist" fish, these vibrations are relayed to the ear through the Weberian ossicle[13,14]. Finally, the lateral line, which senses water flow using hair cells across the surface of the body[3], is also sensitive to low-frequency sounds[15]. Discerning the precise contributions made by each of these organs has been complicated by the nature of sound as it travels through water. Since fish and other aquatic animals generally have a density similar to water's, underwater sound travels almost unimpeded through their bodies, which makes it difficult to confine sound to specific sensory organs. As a result, each hearing organ's contributions can only be isolated by silencing or destroying the other hearing organs[16–18], or through selective recordings of afferent nerves[19–21].

Zebrafish larvae are a powerful model system for studying brain-wide neural networks in general, and sensory networks in particular[22]. They also offer a relatively simple auditory system with only two pairs of otoliths (utricular and saccular), and a lateral line for low frequencies[23]. At the larval stage there is no Weberian ossicle, suggesting that the swim bladder does not contribute to hearing, and the third otolith, the lagena, has not yet developed. Nonetheless, larval zebrafish have sensitivity to a wide range of auditory frequencies and an array of different stimulus features, and have brain-wide auditory networks that share features with those in adult fish[24,25] and with subcortical auditory processing in mammals[26–28].

Because of the challenges of stimulating individual auditory organs in aquatic animals, the utricle's and saccule's specific contributions to hearing have remained uncertain. Indeed, since both the vestibular and auditory systems rely on these two pairs of otoliths at this stage of development[29], this uncertainty extends across two sensory modalities, with a consensus that the utricular otolith is used principally for vestibular perception and the saccular principally for audition[30–32].

## Results

**Optical simulation of sound**. Here, we present Bio-Opto-Acoustic (BOA) stimulation, in which optical forces generate vibrations of otoliths to allow precisely controlled stimulation of specific components of the auditory system. By generating naturalistic vibrations directly in the individual organs, we can stimulate them selectively in a way that is physically prohibited using propagating sound waves. During BOA, optical forces are applied using optical traps[33] (OT), which allow precise and non-invasive mechanical interactions. OT have been used for the manipulation of small transparent objects in a number of biological contexts[34–36], most notably in molecular biophysics[37,38].

In this study, we have designed an optical system capable of applying BOA forces in vivo at frequencies ranging from 1 Hz to 1 kHz (Fig. 1a). We used this to vibrate the otoliths of 6 day post-fertilization (dpf) larval zebrafish (Fig. 1b), allowing controlled stimulation of individual auditory organs for the first time in any intact aquatic animal. To study the brain-wide sensory networks downstream of the otoliths, we combined BOA and actual acoustic stimuli with fluorescent calcium imaging of pan-neuronal GCaMP6s[39] in a light-sheet microscope[40]. This provided volumetric brain-wide recordings of the neuronal activity resulting from real and simulated acoustic stimuli at cellular resolution (Fig. 1c, d). This, in turn, allowed the targeted and systematic exploration of the contributions made by individual auditory organs to auditory and vestibular perception in this important model system.

In our experimental setup, acoustic stimuli were presented from behind the animal to minimize the effects of laterality. Because the larvae have a density similar to water, the traveling acoustic waves are predicted to cause their bodies to vibrate with the water (for our stimuli, along the rostro-caudal axis). However, their otoliths, mostly made out of a tightly packed calcium carbonate crystal, are much denser than water, and therefore move less than the rest of the larva. This relative motion between the otoliths and the rest of the fish causes deflection of hair cells in the ear, thus producing a neural signal that feeds into auditory processing circuitry[41]. Therefore, to simulate acoustic stimuli coming from behind the animal, we vibrated the otoliths along rostro-caudal axis. While this was necessary to provide BOA stimuli equivalent to our auditory stimuli, we note that this direction of movement is not optimal for the utricle, which is most attuned to lateral movements[19].

**Modeling and quantification of Bio-Opto-Acoustics**. Using the optical system presented in Fig. 1a (detailed in "Methods" and Supplementary Fig. 1), we applied alternating OT to opposite (rostral and caudal) sides of the otoliths at various frequencies, creating oscillations of the targeted otolith. These vibrations of the otoliths within the fish body are also observed with sound waves (Fig. 2a, b) due to the high density of otoliths. The motion of each otolith under either auditory or BOA stimulation can be expressed through Newton's law: $F = ma$. In the case of auditory stimulation, it is the acceleration of the otolith relative to the body that is relevant, which results in an effective acceleration of $a_{sound} = (1 - \rho_f/\rho_{ot})a$. We see that the high mass density of the otolith $\rho_{ot}$ compared to the ear fluid $\rho_f$ is essential for high sensitivity. Optical forces can replicate acoustic stimulation by generating similar accelerations of the otolith.

When solving Newton's law equation in the Fourier domain, the solution of the otolith position can be expressed as (details in Supplementary Information):

$$x(f) = \frac{F(f)}{k + i\,2\,\pi\gamma f} \tag{1}$$

Where $\gamma$ is the viscous drag coefficient of ear fluid, $k$ the elasticity constant of hair cells, and where the total force $F$ can include optical force, acoustic sounds, and background forces such as the heartbeat, blood flow, muscle movement, and thermal fluctuations. At low frequencies we simply expect $x \sim F/k$ limited by the hair cell stiffness. At higher frequency, we anticipate the oscillation amplitude to scale as $1/f$.

To quantify the amplitudes of these vibrations at different frequencies, we measured the otoliths' displacement during BOA stimulation (see "Methods" and Supplementary Figs. 2 and 3). To do this, we visualized and quantified the otoliths' vibrations from the spectrograms of their displacements (Fig. 2c). While displacements at low frequencies (1 Hz) are masked by other movements in the animal (heartbeat, blood flow, and voluntary movements), displacements at higher frequencies can be identified and quantified (red circles in Fig. 2c). As clear lines appear for the exact frequency of stimulation, the movement of the otoliths induced by BOA stimulation appears sinusoidal for

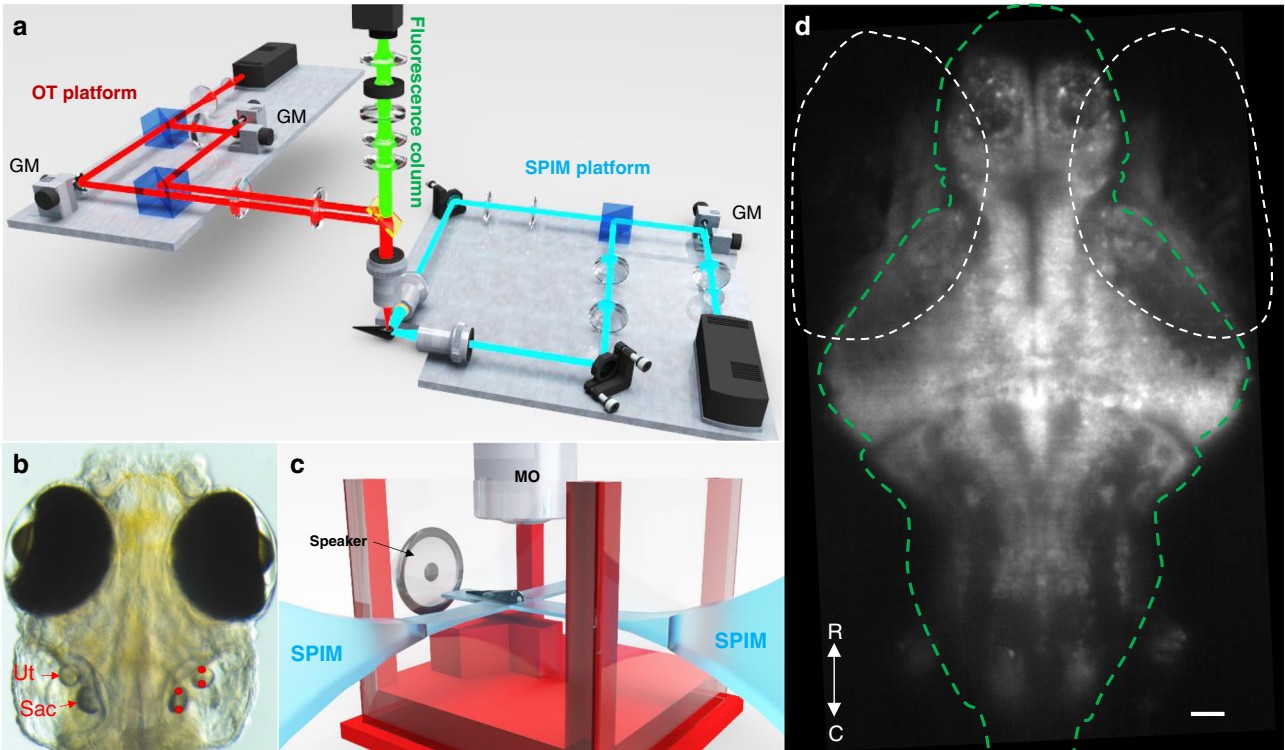

**Fig. 1 Optical and sound stimulation of the auditory system in zebrafish. a** Optical system comprising an OT platform for the generation of two optical traps, a SPIM platform for the illumination of a single plane of zebrafish brain, and a fluorescence column for the imaging of GCaMP6s emissions. Details can be found in the "Methods" section. **b** OT (red dots, right) were placed at two locations within the utricular (Ut) and saccular (Sac) otoliths. The galvo mirrors (GM) on the OT platform displaced each trap from one location to the other at a frequency ranging from 1 Hz to 1 kHz. **c** Sketch representing the placement of a larva in a custom-built-chamber, the SPIM planes, the microscope objective (MO), and the location of the speaker. **d** Example of a fluorescence image recorded from one plane in one fish. The white dashed ovals indicate the eyes, and the green line delineates the brain. R, rostral; C, caudal. Scale bar indicates 10 μm.

those frequencies, as would occur during natural sound stimulation. We also verified that the non-stimulated otolith, located within the same ear, was not significantly moved during the stimulation (Supplementary Fig. 2).

The results confirm the displacement of otoliths using BOA stimulation in vivo. At 10 Hz, we measured displacements of roughly 140 nm, while at 400 Hz, those displacements decreased to about 50 nm for the saccular otolith and 15 nm for the utricular otolith (Fig. 2d) with 400 mW laser power. As expected from Eq. (1) the amplitude of motion decreased with higher frequency. This is unsurprising, since temporally shorter OT forces should produce smaller displacements of the otolith. We found larger displacements across the frequency range for the saccular otolith (Fig. 2d), in spite of its larger size and mass, likely owing to a more favorable geometry that produces stronger trapping forces during BOA stimulation.

The range of frequencies in our BOA stimulus train (Supplementary Fig. 4a) include those associated with vestibular stimuli (1 Hz), those at the interface of vestibular and auditory stimuli (10 Hz), and those that are considered to be in the auditory range (100–1000 Hz). As such, we can flexibly test specific otoliths for their ability to detect and relay information to the vestibular and auditory systems in a way that is not possible with real-world auditory and vestibular stimuli, observing the resulting sensory responses by combining our BOA stimulation with whole-brain calcium imaging in stationary larvae.

**Brain responses to Bio-Opto-Acoustics.** Our modified microscope (Fig. 1) was used to perform brain-wide volumetric GCaMP

imaging (using the *elav3:H2B-GCaMP6s* transgenic line[39], expressing GCaMP6s in the nuclei of all neurons) at 4 Hz volumetric imaging rate (details in "Methods"). This approach was used to map brain-wide responses to a range of BOA frequencies, along with 100 Hz acoustic tones. An example fluorescence image is shown in Fig. 1d. We first automatically segmented regions of interest (ROIs) generally corresponding to individual neurons across the brain, and then extracted signals through time for each ROI, using CaImAn package[42], as stimuli were presented (see "Methods"). We then performed a linear regression to identify ROIs responsive to the acoustic tones and BOA stimuli, followed by a k-means clustering to identify classes of ROIs (clusters) with distinct response profiles to the stimuli. Following this step, we selected clusters responsive to acoustic tones that were also consistently represented across all six larvae tested (see Supplementary Fig. 4 and the selection criteria detailed in the "Methods"). Finally, we warped the 3D structures of all six animals' brains onto one another and onto the Z-brain atlas of the larval zebrafish brain[43], providing a registered reference brain for our responses, and mapped each responsive ROI back to its 3D position within the brain (see "Methods"). This approach allowed us to identify and locate all ROIs responsive to tones or BOA stimulation, and to register them within a common reference.

Our goal was to identify clusters responsive to tones and to compare our targeted BOA stimulation of the utricle, the saccule, or both to actual acoustic stimulation that would affect both otoliths. To remove the complication arising from unilateral versus bilateral stimulation, we pierced the left ear of each larva, which led to a collapse of the otic capsule and a displacement of

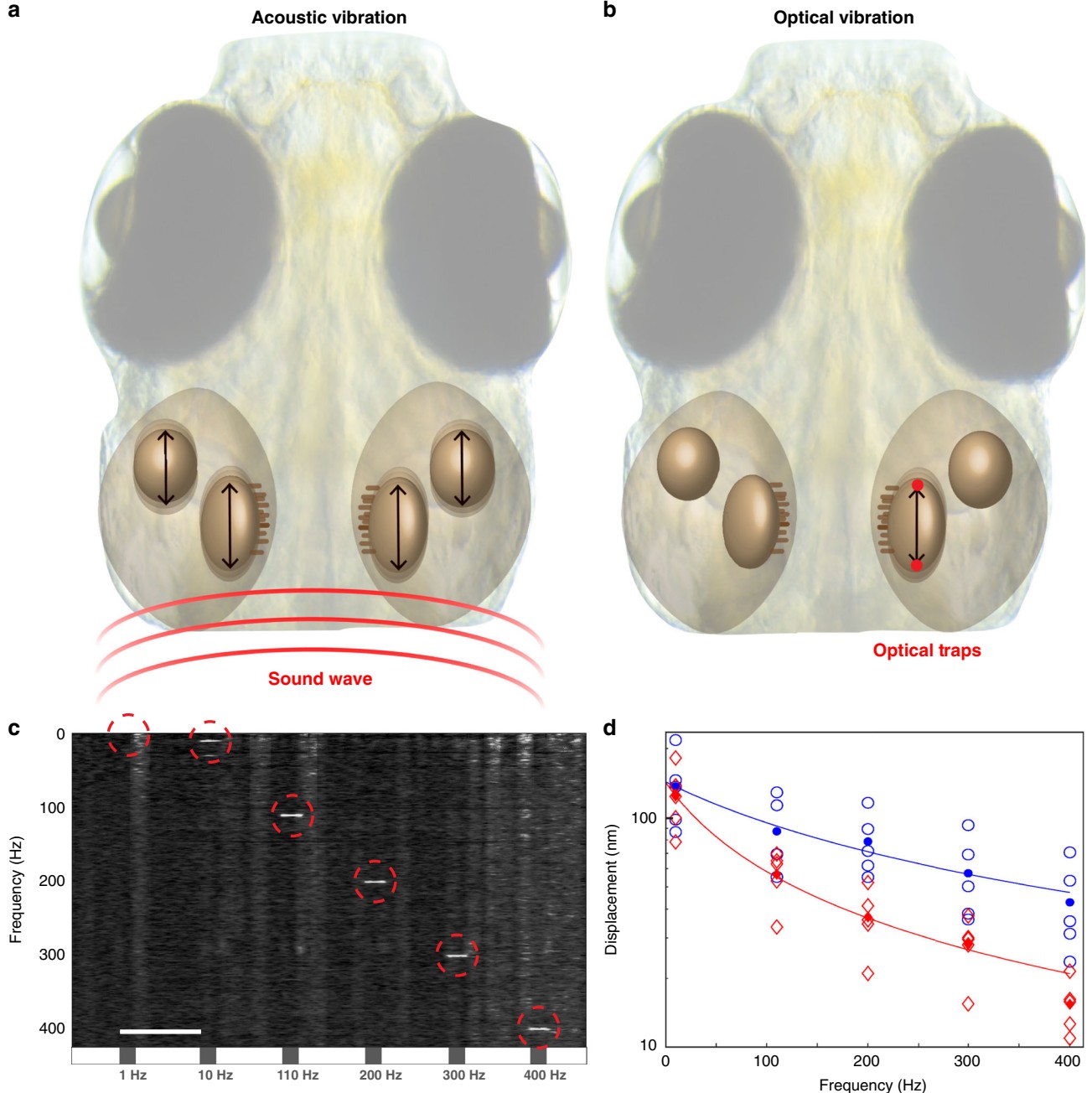

**Fig. 2 Mechanics of sound perception in zebrafish. a** Schematic illustration of sound propagation from a speaker and the resulting movements of the otoliths in larval zebrafish ear. **b** Illustration of OT targeted to alternating sides of one otolith at high speed, and the resulting selective vibration of the targeted otolith. **c** Average spectrogram (normalized over frequency) of the position of the optically manipulated otoliths (both saccule and utricle combined) over time across 5 fish. Gray boxes on the timeline represent 1 s of OT stimulation. The number written under the box represents the OT frequency of the stimulation. See Supplementary Figs. 2 and 3 for more details on the movements of each type of otolith. Scale bar in **c** is 5 s. Diffuse vertical bands are artefacts produced by animal movements, and movements of the otolith at 1 Hz stimulation are masked by background movements. **d** Individual measurements of otolith displacements at different frequencies of BOA stimulation (empty circles and diamonds), displayed on a logarithmic scale. Saccule data are represented in blue and utricle in red. Filled circles and diamonds represent mean values. Fit was performed to Eq. (1), with fitting parameters describing $\gamma$ and $k$ and neglecting mass. $N = 5$ fish.

the otoliths, presumably rendering the ear deaf to auditory stimulation. This ensured comparable stimulation of the right ear only as we applied both BOA and acoustic stimuli at 100 Hz.

We observed that the clusters responsive to 100 Hz tones from the speaker were also responsive to 100 Hz OT stimulation, confirming that BOA stimulation taps into natural auditory circuits in the brain (Fig. 3a). Interestingly, cluster 1 shows that the simultaneous trapping of the utricle and saccule enhances the

neuronal response in a super-additive manner (Fig. 3b). This suggests that the utricle contributes to the detection of a wide range of frequencies. We additionally found a saccule-specific category of ROIs (cluster 2). These ROIs show no pronounced preference for higher frequencies, responding across the range of 1–100 Hz (Fig. 3c), which surprisingly suggests the involvement of the saccular otolith in the detection of low frequencies that are more vestibular than auditory. These data indicate that both the

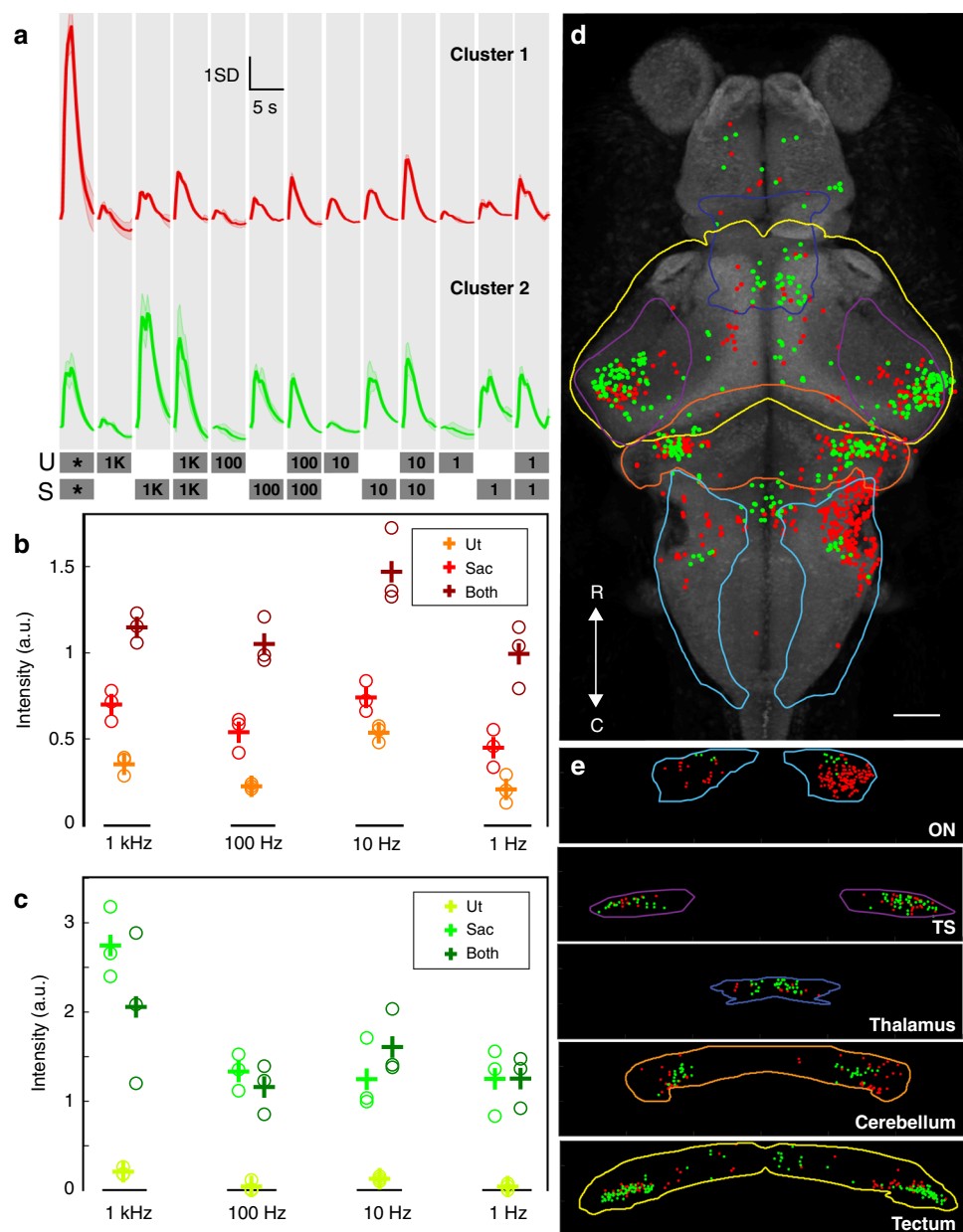

**Fig. 3 Auditory and BOA responses across the brain. a** The average profiles of two 100 Hz tone responsive clusters during acoustic and BOA stimulation. The two bottom lines detail the stimulus train. Gray boxes specify the stimulus windows (1 s of stimulation and 4 s of rest) and the otolith targeted (U, utricle; S, saccule). Numbers on the gray boxes specify the frequency of BOA stimulation in Hz. * represents a 100 Hz auditory tone from a speaker. **b**, **c** Intensity of responses of cluster 1 (**b**) and cluster 2 (**c**) to BOA on the utricle (Ut), saccule (Sac), and both at various frequencies. Circles represent peak responses in individual trials (three in total for each OT configuration), with mean values shown by "+". **d** Locations of ROIs belonging to two functional clusters of auditory neurons, viewed dorsally (red for cluster 1 and green for cluster 2). Brain regions of interest are outlined with colored lines, with light blue for octavolateralis nucleus (ON), purple for torus semicircularis (TS), dark blue for thalamus, orange for cerebellum, and yellow for tectum. Data are from six fish. Scale bars, 50 μm. R, rostral; C, caudal. **e** Distributions of each cluster in the brain regions outlined in **d**, rotated to produce a coronal view.

utricle and the saccule contribute to detecting a wide range of frequencies rather than processing distinct frequency ranges in parallel, and further suggest that auditory and vestibular perception in larval zebrafish are mediated by both otoliths in concert rather than each being carried out by a designated otolith.

In terms of their distributions across the brain (Fig. 3d), these responses occur in structures of the primary auditory pathway, including the octavolateralis nucleus (ON) in the hindbrain, the torus semicircularis (TS) in the midbrain, and the dorsal thalamus in the forebrain (Fig. 3e). These responses are similar

to what has been previously found in auditory studies with zebrafish larvae, which, along with matching to the acoustic stimuli in our own study, corroborates the efficacy of the BOA as a proxy for acoustic stimulation[26,27].

The ON is the first relay center in the brain for auditory and vestibular information[44], and given that the left ear is inactivated and BOA is applied to the right ear, responses to both auditory and BOA stimulation are preferentially found in the right ON (Fig. 3d, e). Activity in the contralateral (left) ON is presumed to result from connections between the left and right ON, as have

been observed for auditory responses in other fish species[45,46] and for lateral line information in zebrafish[47]. We note that this pronounced asymmetry is lost in subsequent processing centers, suggesting that later stages of auditory and vestibular processing are performed bilaterally in this system.

Apart from the structures of the main ascending auditory pathway, the BOA stimulation also elicits responses in the cerebellum and tectum (Fig. 3e). The activity of the cerebellum could be modulating the auditory responses[48], or could be related to motor responses[26]. On the other hand, the tectum is known to integrate multiple sensory modalities, including auditory information[49]. It is possible that auditory information in the tectum can be used to generate a spatial map, as the homologous superior colliculus does in mammals[50].

Finally, we note a transition from the ON where we find a preponderance of cluster 1 ROIs, to brain regions later in the auditory pathway where we see more balance between the clusters or a greater number of cluster 2 ROIs. This suggests a possible shift from general responses in the ON (with responses to all tested frequencies and drawn from both otoliths) to more selective responses (favoring high frequencies, and with selective control by the saccular otolith) later in the auditory processing pathway.

## Discussion
In this study, we have introduced BOA as a method for the precise stimulation of the auditory system with light. Our observations, while they provide unambiguous accounting of saccular and utricular contributions to brain-wide auditory processing, likely understate the breadth and complexity of these sensory pathways. Further analyses with bilateral BOA stimulation, a greater number of stimulus frequencies, and a correspondingly greater diversity in functional clusters, will be necessary fully to map the nuances of each otolith's contribution to auditory and vestibular processing. Such studies will allow the detailed characterization of frequency discrimination, analyses of the convergence and differentiation of vestibular and auditory pathways in the brain, and the ontogeny of auditory and vestibular processing during development. As these structures are part of the main ascending auditory pathway of teleost[48,51], which are shared with amphibians and mammals[48,52–54], BOA represents an effective tool for the exploration of the vertebrate auditory system.

## Methods
**Animals**. All procedures were performed with approval from The University of Queensland Animal Welfare Unit (in accordance with approval SBMS/378/16). Zebrafish (*Danio rerio*) larvae, of either sex, were maintained at 28.5 °C on a 14 h ON/10 h OFF light cycle. Adult fish were maintained, fed, and mated as previously described[55]. All experiments were carried out in nacre mutant *elavl3:H2B-GCaMP6s* larvae[39] of the TL strain.

**Sample preparation**. Zebrafish larvae at 5 days post-fertilization (dpf) were immobilized in 2% low melting point agarose (LMA) (Sigma-Aldrich) on microscope slides. Using the thin sharp end of a pulled pipette, the left ears of zebrafish larvae were pierced. The fish were released from LMA and placed back in their incubator. The following day, these larvae (6 dpf) were immobilized dorsal side up in 2% LMA on microscope slides. Each embedded fish was transferred to custom made, 3D printed chamber[56], which was filled with E3 media[55]. Larvae were then allowed to acclimate for 15 min prior to imaging on the custom-built dual optical trapping microscope presented in Fig. 1.

The protocol for the experiments can be found in ref. [57].

**Dual optical trapping system and targeting**. The dual optical trapping (OT) system (Fig. 1 and Supplementary Fig. 1) was composed of an infrared laser (1070 nm IPG Photonics YLD-5 fiber laser), a half-wave plate (HWP) that rotates polarization by 45°, and a polarizing beamsplitter (PBS) that splits the incoming beam into two beams of the same intensity with perpendicular polarizations. The two independent beams (Trap 1 for the utricle and Trap 2 for saccule) were

reflected off of a galvo mirror (GM) (Thorlabs GVSM002/M). The two beams were recombined with a second polarizing beamsplitter and a telescope (L2 = 150 mm and L3 = 300 mm focal length). An additional lens L1 (150 mm focal length) was added into Trap 2 path to displace this trap +20 μm in Z (above Trap 1) in order to reach the saccule, located around 20 μm above the utricle. The beams were then reflected off a 950 nm cut-off wavelength shortpass dichroic mirror (DM) in the imaging column, and projected onto the back focal plane of a ×20 1NA Olympus microscope objective (XLUMPLFLN-W). This created two tightly focused spots at the imaging plane, or +20 microns above the imaging plane of the microscope objective. The positions $(x, y)$ of each pair of optical traps (two trap positions for each otolith) were steered with the galvo mirrors. The two galvo mirrors were driven with Arduinos (Leonardo) in order to place the OT beam at precise $(x, y)$ locations, and oscillated between these predetermined locations at variable frequencies (1, 10, 100, and 1000 Hz). Two shutters (Thorlabs SHB1T) allowed independent gating of the OT and were also driven using an additional Arduino (Leonardo). A laser power of 400 mW was used and gauged using a power meter at the focal plane of the ×20 1NA objective. From our previous study on force measurements within otoliths[58], we know that traps positioned between 1 and 3 μm from the edge of the otolith produce the largest forces, and that this force is radial. Therefore, we positioned each trap in the rostral-caudal axis of each otolith (utricular and saccular otolith) and about 2 μm from its edge. Once the trapping positions were determined, we performed the BOA stimulation and calcium imaging simultaneously.

**Acoustic stimulation**. Acoustic stimulation was provided by a mini speaker (Dayton Audio DAEX-9-4SM Skinny Mini Exciter Audio and Haptic Item Number 295-256) wired to an amplifier (Dayton Audio DA30 2 × 15W Class D Bridgeable Mini Amplifier). The mini speaker was glued to the glass coverslip wall of the 3D printed chamber located behind the fish. The sound intensity level was selected as the average between the sound intensity obtained when a fish would start to detect the 100 Hz tone (GCamp6s fluorescent activity seen in the ON), and the sound intensity obtained when a fish would start to have escape responses to 100 Hz tones. Five fish were observed to determine this parameter. This parameter was kept constant for all acoustic stimulation experiments.

**Imaging and analysis of otolith movements**. In order to image both the utricular and saccular otoliths at high speed (1 kHz), we built a system to provide transillumination of the fish with a bright white LED light under the specimen. Using μManager[59], we cropped the video recordings to a tight region around each otolith to allow the acquisition frame rate to reach 1 kHz. Each otolith was recorded separately, and otolith motion was estimated from recorded movies using an efficient subpixel image registration by cross-correlation method[60]. Using this method, we calculated the displacement of each otolith in X and Y in response to the optical manipulation. Since the resulting traces were noisy (subpixel movements), we represented the data in the Fourier domain using a spectrogram (Fig. 2 and Supplementary Figs. 2 and 3) with a 1 s window. The choice of the window time was optimized to allow maximum detection of the frequency within the displacement, and as a consequence, the displacement of the otolith over time appears to happen during more than 1 s of the stimuli (Supplemental Fig. 3b, c).

It is worth noting that during BOA stimulation, the camera detects a very small fraction of the laser light used for the trap, and as a consequence, the raw signal contains a small step function when the stimulus appears and disappears. This results in the presence of multiple frequencies (appearing as vertical lines on the onset and offset times of the traps) in the spectrogram with weak intensities.

**Fluorescence imaging system**. Calcium imaging and OT were performed through the same ×20 objective. For the fluorescence imaging of a chosen depth on the PCO edge 5.5 camera, we used a combination of a filter (Thorlabs FF01-517/520-25), tube lens (TL = 180 mm focal length, Thorlabs AC508-180-A), relay lenses (RL, Thorlabs AC254-125-A-ML), ETL (Optotune EL-10-30-Ci-VIS-LD driven with Gardasoft TR-CL180), and offset lens (OL, Eksma Optics 112-0127E). The scanning light sheet was generated using a 488 nm laser (OBIS 488lx), scanned with 2D galvo mirrors (GM, Thorlabs GVSM002/M), a 50/50 beamsplitter, and a 1D line diffuser (1D-LD, RPC Photonics EDL-20-07337). Details on the optical path for the whole system can be found in Supplementary Fig. 1. Further details on the use of the ETL can be found in ref. [61] and on the use of the 1D diffuser in ref. [40].

With this configuration, we were able to scan 250 μm of brain tissue above the original imaging plane, where utricle otolith is placed. One galvo scanning direction (Y) created the light sheet while the second direction (Z) created the depth scan in the sample. The two mirrors were driven independently using Arduinos (DUE) with custom-written code. The Y scanning was a sawtooth scan at 600 Hz, which was synchronized to the camera acquisition to ensure similar illumination for each camera acquisition. The Z galvo was driven in 10 ms steps to scan the light sheet in Z through the sample. The 50/50 beamsplitter created two light sheets, one projecting into the rostral side and one into the right side of the fish. The 1D line diffuser was placed just after the galvos to reduce shadowing effects in the planes[40]. The imaging system was controlled using μManager, based on ImageJ[62,63]. In our experiments, an exposure time of 10 ms was chosen for each plane during volumetric imaging, with laser power output of 60 mW, which was

attenuated to 1.5 mW for each plane at the sample. A total depth of 250 μm in Z was scanned[64], with 25 planes at 10 μm intervals, resulting in a 4 Hz volumetric acquisition rate. We commenced laser scanning 30 s prior to imaging neural activity to eliminate responses to the onset of this off-target visual stimulus.

**Extraction of fluorescent traces**. The volumetric scan was first transformed into a hyperstack in Fiji[65], and then separated into individual slices. We used the CaImAn package to analyze our images[66,67] and extract the fluorescent traces of each ROI from every slice (http://github.com/flatironinstitute/CaImAn). We compensated for small movements using a rigid registration[68]. The greedy ROI method was used to initialize, for each slice, 4000 components from which to extract, demix, and denoise the fluorescent traces using an autoregressive model of order 1[66]. We used a correlation threshold of 0.8 to merge overlapping ROIs and avoid over-segmentation. The components were updated before and after the merge steps, empty components were discarded, and the components were ranked for fitness as in ref. [66].

**Whole-brain analysis of fluorescent traces**. The active ROIs and their respective fluorescent traces were further analyzed in MATLAB with a custom-written code. This code and descriptions of its use can be found in the "Code availability" section and previous versions are described in refs. [69,70]. The traces from six fish were pooled and z-scored. Regressors were built for the stimulus train presented with a typical GCaMP response at each stimulus onset (Supplementary Fig. 4a). A linear regression was performed between all the fluorescent traces and the regressors. The coefficient of determination ($r^2$) of the linear regression models was used to select stimulus-responsive ROIs, and we chose a 0.2 threshold based on the $r^2$ distribution of our models to allow for conservative filtering of the data (Supplementary Fig. 4b). The next step was clustering with the k-means method. The fluorescent traces passing the linear regression test were clustered into 120 clusters using k-means with the cityblock distance and five replicates. All the clusters' averages were correlated with the regressors, and the clusters responsive to tones and specific to the optical vibration of the saccule with a response at least 1 SD above baseline were selected. The fluorescence traces within each resulting cluster were compared to the regressors using linear regression and the ROIs showing $r^2$ values above 0.4 were selected (Supplementary Fig. 4c).

Finally, clusters were filtered with the following selection criteria:

(1) Responsivity to each tone stimulus as a GCaMP6s profile,
(2) Responsivity to each tone stimulus with a response above 1 SD to baseline,
(3) Less than 90% of the ROIs within the cluster are represented in each individual fish.

**Spatial registration of fluorescence imaging to a reference brain**. We used Advanced Normalization Tools (ANTs, https://github.com/ANTsX/ANTs) to compute the diffeomorphic map between the time-averaged 3D image stack of each fish and the H2B-RFP reference of Z brain[43,71,72]. The same mapping was used to warp the centroid coordinates for each ROI of interest to the H2B-RFP frame of reference, which includes 294 segmented brain regions[43]. We used MATLAB to represent each ROI centroid as a sphere within the Z-brain reference brain image.

**Reporting summary**. Further information on research design is available in the Nature Research Reporting Summary linked to this article.

## Data availability

The .tif files data that support the findings of this study are available in "Bio-Opto-Acoustics" with the identifier https://doi.org/10.14264/9809ff7. Source data are provided with this paper.

## Code availability

Codes can be downloaded from https://doi.org/10.14264/9809ff[57].

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

## Acknowledgements

Support was provided by an NHMRC Project Grant (APP1066887), a Simons Foundation Pilot Award (399432), a Simons Foundation Research Award (625793), and two ARC Discovery Project Grants (DP140102036 & DP110103612) to E.K.S.; an NHMRC Project Grant (APP1165173) to E.K.S. and H.R.; an ARC Discovery Project Grant (DP180101002) to H.R.; and the Australian National Fabrication Facility (ANFF), QLD node. The research reported in this publication was supported by the National Institute of Neurological Disorders and Stroke of the National Institutes of Health under Award Number R01NS118406. The content is solely the responsibility of the authors and does not necessarily represent the official views of the National Institutes of Health. We thank Nicolas P. Mauranyapin for the design of Fig. 1a. We thank members of the Scott and Rubinsztein-Dunlop groups for discussions of this manuscript.

## Author contributions

I.A.F. and E.K.S. designed the project. I.A.F. and M.A.T. built the optical system. I.A.F. and R.E.P. designed the fish chamber. I.A.F. carried out the experiments. I.A.F., E.M.-L., and G.V. processed the fluorescent neuronal data. I.A.F., M.A.T., E.M.-L., G.V., H.R., and E.K.S. wrote the manuscript.

## Competing interests

The authors declare no competing interests.
