## [Peer Review File · Nature Communications]

Reviewers' Comments:

Reviewer #1:

Remarks to the Author:

The manuscript reports on recent findings concerning the functionality of otoliths in zebrafish enabled by a new method, bio-opto-acoustics (BOA). In BOA, oscillating optical traps generate specific acoustic stimuli in situ, which are investigated by direct monitoring of the auditory response in the brain, mediated by calcium-based fluorescence imaging in a 2-beam light-sheet microscope.

One or more oscillating double-spot optical tweezers were able to induce an acoustic stimulus of selectable frequency, but in a highly localized manner, i.e. close to a chosen otolith. With the ability to selectively stimulate the utricular and the saccular otoliths in various frequency ranges (from 1 Hz to 1 kHz), either individually or together, the approach made it possible to shed some light on the different responses of the two types of otoliths. It could be ruled out that - contrary to the prevalent hypothesis - the two types are 'responsible' for (i.e. sensitive to) disjoint frequency regions. The level of localization and selectivity of the acoustic stimulus that was necessary to conclude on this question cannot be reached by sound from a microphone. The approach is not only more precise than existing methods, but also non-invasive - in contrast to previous routes taken that were based on studying changes brought about by the selective destruction of otoliths.

I only have a few, very minor, requests:

In Fig.3 b) and c), what do the circles stand for? And why is the utricular response so low in c)? Optional suggestion: Does the method also provide new insights into structural differences of the two types of otoliths (utricular and saccular) ? Are they only different in size and shape, or are there also small differences in mass density and/or elasticity (and thus in the mechanical impedance)? It would be nice to get some more information on this, if possible.

Conclusions: In my opinion, this is a very well written paper, and a great example for a combination of tools in a highly innovative and clever way to enable testing of a hypothesis that was hitherto out of reach. I strongly recommend publication (basically 'as is') in Nature Communications.

Reviewer #2:

Remarks to the Author:

In this manuscript, Favre-Bulle and colleagues present an innovative optic-trapping based approach — bio-optic acoustics (BOA) — to stimulate otolithic organs while measuring neural activity. This exciting technique allows the authors to identify clusters of neurons with distinct characteristics in response to "vestibular" and "auditory" stimulation. Taken together, the results are convincing and the manuscript is well-written. However, in our opinion, the manuscript currently lacks one key control, and would additionally benefit from better contextualization with respect to previous studies. There is also room for improvement to better highlight relevant aspects of the data. Please see below for specific comments.

Yunlu Zhu, Ph.D. and David Schoppik, Ph.D.

Major comments

1. The framing of the manuscript rests on the assertion that the optical trapping approach singularly activates one particular end-organ (i.e. "first unambiguous accounting," 180. However, the manuscript does not demonstrate that this is the case. The manuscript needs evidence that moving one otolith does not displace the other. The authors presumably have this data from

experiments where they have trapped both the utricle and saccule in the same ear, where they could read off the consequences of moving one on the other. This is no trivial matter: moving an otolith in the closed system of the otic capsule certainly has the potential to displace endolymph and, by extension, the other otolith. Alternatively, the manuscript needs considerable reframing.

2. Notably, there are many counterexamples to the claims in the manuscript of "distinct and non-overlapping roles for the different otoliths." For example, see Allen, Blaxter & Denton '76 for herring, or Denton, Gray & Baxter 1979 where both the utricle and lagena are thought to participate in hearing. The references cited in the manuscript should have covered this, but one additional point of reference is Hawkins "Underwater Sound and Fish Behaviour" in the book, *The Behaviour of Teleost Fishes* (ed. Pitcher).

3. The manuscript would benefit from discussion of the choice of axis of stimulation relative to the orientation of the hair cells, particularly since (unless we've misunderstood the diagram) the axis of choice for the saccule is orthogonal to the primary mechanical axis for most hair cells.

4. Line 67. The manuscript claims that "The bodies of fish, which have a density close to water, move in concert with traveling waves as sound passes through them". However, the statement is not completely true because the inner ear of adult fish also receives sound pressure originating from the gas-filled swim bladder (Lu 2004, Popper and Fay, 2011). Although the swim bladder contribution requires the Weberian ossicles which do not appear until later developmental stages (Higgs et al., 2003), this additional sound transmission mechanism needs to be discussed here in order to make the statement more accurate.

> Higgs DM, Rollo AK, Souza MJ, Popper AN. Development of form and function in peripheral auditory structures of the zebrafish (*Danio rerio*). *J Acoust Soc Am*. 2003 Feb; 113(2):1145-54.

> Lu Z. Neural mechanisms of hearing in fishes. In: von der Emde G, Mogdam J, Kapoor BG, editors. *The sense of fish: Adaptations for the reception of natural stimuli*. New Delhi: Narosa Publishing House; 2004. pp. 147-172.

> Popper AN, Fay RR. *Hear Res*. Rethinking sound detection by fishes. 2011 Mar; 273(1-2):25-36.

5. In addition, the manuscript would benefit from discussion of the development of hearing in zebrafish and why 5 dpf larvae were chosen for the study.

6. Figure 2. The otoliths' displacements during BOA stimulation are around 50-140 nm. How does that compare to the effect of acoustic stimulation? Approximately, what are the equivalent sound pressure levels? If the authors already have the data (or can refer to previous studies), they should be discussed in the manuscript. We note that these additional measurements are not necessary for publication.

7. Supplemental Figure 1a & b. The authors demonstrate the motion of otoliths in individual fish under BOA stimulation using spectrograms. We think these figures are important and the results are clear. However, we found it difficult to spot red signals on the black background. We would suggest presenting these results differently to highlight the relevant data. Please also add annotations representing stimulation windows.

8. Supplemental Figure 1a. The saccular otolith seems to show a short motion signal at 250/300 Hz time-locked to each stimulus particularly at lower BOA frequencies. What could be the cause of these signals? Is this common across different fish?

9. Supplemental Figure 1c & d. In all panels, the time axes begin at 0s but stimulations start around 0.5s. We think visualization of the results can be improved by aligning the beginning of the stimulus at time 0 and/or highlight the 1-s stimulation window.

10. Line 282-286. These results refer to Supplementary Figure 2a and 2b instead of 1a and 1b.

11. Figure 3d. The manuscript should describe which color represents which cluster in 3d.

12. Because this is the first time BOA is presented the manuscript would benefit from clarity in exactly what the stimulation does, much of which is glossed over. For example, in Figure 2c why is there a lag at 1Hz? It appears that there is differential activity in time following stimulation at high frequency? Why? Is it true that a higher freq stim produces longer lasting consequences? Finally, are the data from the utricle and saccule combined for this figure? Please assay whether or not they are sufficiently similar to merit. We encourage the editors to allow the authors the space to fully flesh out the relevant details.

Minor comments

1. Mixed uses of "Figure" and "Fig."
2. Mixed uses of "Supplementary" and "Sup"
3. Line 110. Typo "Figure 2.a"
4. Line 142. Typo "Fig. 3.a"
5. claim in 35-37 this could use a citation
8. line 136-137 how do you know it is deaf? Similarly, line 20 how do you know how the stimulus is perceived?

Reviewer #3:

Remarks to the Author:

Favre-Bulle et al. describe an interesting experimental approach to stimulate the auditory system of zebrafish larvae with optical forces. They use their technique to characterize associated neural activity and the role of the utricular and saccular otoliths.

In the abstract the authors say that they use "optical forces to generate localized sound in vivo", and I was trying to imagine how they would do so in water, in the vicinity of the fish. As it turns out, the authors use an optical trap to vibrate the otoliths of larval zebrafish, allowing controlled stimulation of the auditory organ. In the following the authors compare this stimulation to "true sound" from a speaker. I think it is misleading to distinguish between different kinds of sound in this case. Direct stimulation by vibrating the otoliths is not localized generation of sound. This needs to be clarified.

I am also missing a description of how the oscillation and displacement of the otoliths are induced. There seem to be two alternating optical traps at a given distance. How is this distance determined and what influence does this have on the performance of the oscillator? It is not intuitively clear what this optimal distance should be. Since two shutters are used I assume that the waveform of the frequency is rectangular. Does this result in a sinusoidal oscillation? Is the exact waveform of the oscillation relevant in trying to simulate stimulation by sound waves?

"These vibrations of the otoliths relative to the fish body are similar to what a sound wave would produce". I cannot find a control experiment that demonstrates that indeed the vibrations induced by the alternating optical trap are similar to what is observed when the ear is hit by a sound wave. How is the frequency response under these natural conditions (relative displacement vs. frequency)? Which direction do the otoliths move relative to the fish? Is this the same direction as you induce with the optical trap?

The figures are of low quality, the graphs are hard to read, and the drawings in Fig. 1 and 2 are very qualitative and not precise enough. Where are the red beams pointing in 2b? Is this to scale? This is not how focused laser beams move when scanned in the BFP. I would also like to see a more detailed optical layout. The 3D rendering is not helpful and does not show the ray paths.

Overall, the manuscript presents an interesting idea and a nice implementation, however, I am missing a lot of experimental details and measurements that convince me that the stimulation by the optical traps is similar to the response to real sound waves. If this is meant to be a methods paper, more emphasis needs to be put on the justification and characterization of the experimental setup and procedures.

Reviewer #1:

The manuscript reports on recent findings concerning the functionality of otoliths in zebrafish enabled by a new method, bio-opto-acoustics (BOA). In BOA, oscillating optical traps generate specific acoustic stimuli in situ, which are investigated by direct monitoring of the auditory response in the brain, mediated by calcium-based fluorescence imaging in a 2-beam light-sheet microscope.

One or more oscillating double-spot optical tweezers were able to induce an acoustic stimulus of selectable frequency, but in a highly localized manner, i.e. close to a chosen otolith. With the ability to selectively stimulate the utricular and the saccular otoliths in various frequency ranges (from 1 Hz to 1 kHz), either individually or together, the approach made it possible to shed some light on the different responses of the two types of otoliths. It could be ruled out that - contrary to the prevalent hypothesis - the two types are 'responsible' for (i.e. sensitive to) disjoint frequency regions. The level of localization and selectivity of the acoustic stimulus that was necessary to conclude on this question cannot be reached by sound from a microphone. The approach is not only more precise than existing methods, but also non-invasive - in contrast to previous routes taken that were based on studying changes brought about by the selective destruction of otoliths.

I only have a few, very minor, requests:

1. In Fig.3 b) and c), what do the circles stand for? And why is the utricular response so low in c)?

The circles represent the average peak responses across ROIs belonging to the relevant cluster, within an individual trial. The averages for all three trials are shown by the "+" for each cluster and treatment. We have changed the wording of the Fig 3 legend to make this clearer (lines 569-570). We have also removed the "+" showing the mathematical sum of the two individual stimuli, since this figure is more focussed on the repeatability of the method than on facilitation between the two otoliths' signals (which would be the reason for comparing a mathematical sum to the response of the dual trap). Incidentally, we have also offset the data for the three treatments slightly to allow them to be more easily assessed visually (Figure 3b and c).

In c), the low values for responses to utricular stimulation reflect the lack of responses seen in cluster 2 ROIs during utricular-alone stimulation (consistent with the responses shown in panel 3a).

2. Optional suggestion: Does the method also provide new insights into structural differences of the two types of otoliths (utricular and saccular)? Are they only different in size and shape, or are there also small differences in mass density and/or elasticity (and thus in the mechanical impedance)? It would be nice to get some more information on this, if possible.

It is true that all of these factors would affect the relationship between the OT and the movements of the otoliths, and we agree that this is an interesting, if complex, space. Since we have no clear way of separately analyzing the effects of density, mass, shape, refractive index, and mechanical impedance, we have not attempted to address their interactions. Since this study is aimed at demonstrating the selective vibration of these targets (see Reviewer 2, question 1), and the resulting sensory activity, we believe that this is beyond the scope of the current study.

Conclusions: In my opinion, this is a very well written paper, and a great example for a combination of tools in a highly innovative and clever way to enable testing of a hypothesis that was hitherto out of reach. I strongly recommend publication (basically 'as is') in Nature Communications.

Reviewer #2 (Remarks to the Author):

In this manuscript, Favre-Bulle and colleagues present an innovative optic-trapping based approach — bio-optic acoustics (BOA) — to stimulate otolithic organs while measuring neural activity. This exciting technique allows the authors to identify clusters of neurons with distinct characteristics in response to “vestibular” and “auditory” stimulation. Taken together, the results are convincing and the manuscript is well-written. However, in our opinion, the manuscript currently lacks one key control, and would additionally benefit from better contextualization with respect to previous studies. There is also room for improvement to better highlight relevant aspects of the data. Please see below for specific comments.

Yunlu Zhu, Ph.D. and David Schoppik, Ph.D.

Major comments

1. The framing of the manuscript rests on the assertion that the optical trapping approach singularly activates one particular end-organ (i.e. “first unambiguous accounting,” 180. However, the manuscript does not demonstrate that this is the case. The manuscript needs evidence that moving one otolith does not displace the other. The authors presumably have this data from experiments where they have trapped both the utricle and saccule in the same ear, where they could read off the consequences of moving one on the other. This is no trivial matter: moving an otolith in the closed system of the otic capsule certainly has the potential to displace endolymph and, by extension, the other otolith. Alternatively, the manuscript needs considerable reframing.

This is, indeed, an important part of what we claim to be doing in this study. We did not have such data from our original dataset, so we have done new experiments where we measured the displacement of both otoliths, in both x- and y-directions, while optically manipulating only the utricular otolith in the x direction (as in the rest of the study). These results are now shown in Supplemental Figure 2, and also copied below. A reference to these results has been added to the manuscript in lines 115-116. This provides evidence that targeting the utricular otolith causes marked movement in the expected (X) direction without appreciably moving the saccular otolith in the same ear.

Supplemental Figure 2. Otoliths displacements under the BOA stimulation. a-e. Measurement of displacements of utricle (red) and saccule (blue) otoliths in X and Y directions (same fish, same ear, same stimuli train, different times) when the utricular otolith undergoes BOA stimulation in the X direction. Hollow circles represent data points for each fish. Filled circles represent the mean value across fish. N=3 fish.

2. Notably, there are many counterexamples to the claims in the manuscript of “distinct and non-overlapping roles for the different otoliths.” For example, see Allen, Blaxter & Denton ’76 for herring, or Denton, Gray & Baxter 1979 where both the utricle and lagena are thought to participate in hearing. The references cited in the manuscript should have covered this, but one additional point of reference is Hawkins “Underwater Sound and Fish Behaviour” in the book, *The Behaviour of Teleost Fishes* (ed. Pitcher).

We thank the reviewers for this additional information. In our original manuscript, we were focussing on the state of play in the larval zebrafish literature, but we did not make this clear, and probably should have approached the literature more broadly in the first place. We have addressed these shortcomings in the revised manuscript in lines 34-61, which represent a fundamental rewriting of our introduction in response to these and other reviewers. The suggested references have been added.

3. The manuscript would benefit from discussion of the choice of axis of stimulation relative to the orientation of the hair cells, particularly since (unless we’ve misunderstood the diagram) the axis of choice for the saccule is orthogonal to the primary mechanical axis for most hair cells.

This is a good point. Since we are endeavouring to match real auditory and BOA stimulation, we restricted the direction of BOA stimulations to the one (rostral-caudal) axis that we have stimulated with our speaker. We believe that our diagrams led to the reviewers’ confusion. Our original Figure 2, which was intended to be conceptual, showed horizontal shaking of the otoliths in a context

where this would be interpreted as being a lateral movement. This was misleading. We have completely reworked Figure 2 to make it clear that in both cases, the movement is rostral-caudal.

We have also made additions to the manuscript highlighting the possible impacts of motion across this axis on our results, particularly for the utricular otolith. These additions can be found in lines 80-90 and 92.

4. Line 67. The manuscript claims that "The bodies of fish, which have a density close to water, move in concert with traveling waves as sound passes through them". However, the statement is not completely true because the inner ear of adult fish also receives sound pressure originating from the gas-filled swim bladder (Lu 2004, Popper and Fay, 2011). Although the swim bladder contribution requires the Weberian ossicles which do not appear until later developmental stages (Higgs et al., 2003), this additional sound transmission mechanism needs to be discussed here in order to make the statement more accurate.

> Higgs DM, Rollo AK, Souza MJ, Popper AN. Development of form and function in peripheral auditory structures of the zebrafish (*Danio rerio*). *J Acoust Soc Am*. 2003 Feb; 113(2):1145-54.

> Lu Z. Neural mechanisms of hearing in fishes. In *The Senses of Fish*. (Springer), pp. 147-172. In: von der Emde G, Mogdam J, Kapoor BG, editors. *The sense of fish: Adaptations for the reception of natural stimuli*. New Delhi: Narosa Publishing House; 2004. pp. 147-172.

> Popper AN, Fay RR. *Hear Res*. Rethinking sound detection by fishes. 2011 Mar; 273(1-2):25-36.

Again, our original writing was larval zebrafish-centric. We now address these complexities in a revised introductory paragraph, found in lines 34-61. The requested references have been added.

5. In addition, the manuscript would benefit from discussion of the development of hearing in zebrafish and why 5 dpf larvae were chosen for the study.

We apologize that this was misstated in the original manuscript's methods. These experiments were done on 6dpf larvae, as we now state both in the main text (line 71) and in the methods (218). This is a frequently-used age for such imaging studies due to trade-offs between the animals' size and complexity versus their suitability for whole-brain imaging. We have not added a justification for using this particular age, but we have added material discussing what is known about the larval zebrafish hearing system at this stage of development (lines 51-56).

6. Figure 2. The otoliths' displacements during BOA stimulation are around 50-140 nm. How does that compare to the effect of acoustic stimulation? Approximately, what are the equivalent sound pressure levels? If the authors already have the data (or can refer to previous studies), they should be discussed in the manuscript. We note that these additional measurements are not necessary for publication.

Because of the strong vibration in the animal as a whole during acoustic stimulation, we have found these measurements (relative movements of the body versus the otoliths) very difficult to make. We have made a concerted effort to gather such data, but the vibrations of the body (a small part of which occupies the whole field of view while taking such measurements at high magnification) have proven too difficult to quantify in our hands.

7. Supplemental Figure 1a & b. The authors demonstrate the motion of otoliths in individual fish under BOA stimulation using spectrograms. We think these figures are important and the results are clear. However, we found it difficult to spot red signals on the black background. We would suggest presenting these results differently to highlight the relevant data. Please also add annotations representing stimulation windows.

These are very delicate measurements of subpixel movements, and as a consequence, we have very little signal relative to the noise in the system. As such, these data will never look clear, and the figures that we have provided are the best that we are able to generate. We could binarize these spectrograms, but we feel that this would be “hiding” the noisiness of the system, which we feel is inappropriate. As such, we have left the figure (now Supplemental Figure 3) unchanged, but we are happy to revisit this if the reviewers would like.

Stimulus windows have been added on the X axis of all panels for this figure, as requested.

8. Supplemental Figure 1a. The saccular otolith seems to show a short motion signal at 250/300 Hz time-locked to each stimulus particularly at lower BOA frequencies. What could be the cause of these signals? Is this common across different fish?

These vertical lines in the spectrograms seen at the onsets and the offsets of the trap stimuli are artifacts. They result from the choice of the duration of the window used to create the spectrogram (1 sec in this case). The camera picks up a small fraction of the laser light used for the trap, and as a consequence, the raw signal intensity has a small step function when the stimulus appears and disappears. This results in the presence of multiple frequencies in the spectrogram, which is a Fourier transform of one second (in our case) of the signal. We have briefly addressed this in the Methods section (Lines 258-268).

9. Supplemental Figure 1c & d. In all panels, the time axes begin at 0s but stimulations start around 0.5s. We think visualization of the results can be improved by aligning the beginning of the stimulus at time 0 and/or highlight the 1-s stimulation window.

We have added the period of stimulation to this figure (now Supplemental Figure 3). This introduces an apparent rise in the displacement prior to the onset of the stimulus, which is the result of temporal averaging of the signal and smoothing across consecutive windows. We have added an explanation of this to the Methods section (Lines 260-263), and have commented on the effect in the legend of Supplemental Figure 3 (formerly Supplemental Figure 1).

10. Line 282-286. These results refer to Supplementary Figure 2a and 2b instead of 1a and 1b.

Thank you, we have corrected this.

11. Figure 3d. The manuscript should describe which color represents which cluster in 3d.

We have added a sentence to the figure legend stating this (Lines 571-574).

12. Because this is the first time BOA is presented the manuscript would benefit from clarity in exactly what the stimulation does, much of which is glossed over.

a. For example, in Figure 2c why is there a lag at 1Hz?

In Fig 2c at 1Hz, the otolith displacement is not visible due to noise within the larvae. The large diffuse vertical band shortly after this stimulation is a coincidental artifact, likely due to a physical movement from one of the larvae at this time. Other such artifacts are visible at other timepoints. Both of these issues are now addressed in the Fig. 2 legend (Lines 555-556).

b. It appears that there is differential activity in time following stimulation at high frequency? Why? Is it true that a higher freq stim produces longer lasting consequences?

We do not believe that there are differences in the durations of the responses to different frequencies of BOA (although the magnitudes differ). We believe that this question refers to the diffuse vertical lines, which are the same motion artifacts described above. We clarified the motion artifacts in Lines 111-113.

c. Finally, are the data from the utricle and saccule combined for this figure? Please assay whether or not they are sufficiently similar to merit. We encourage the editors to allow the authors the space to fully flesh out the relevant details.

Yes. These data combine utricular and saccular data (now stated more clearly in lines 551-552). We have found similar patterns across these two otoliths, although the saccular otolith usually moves more than the utricular otolith. This is quantified in Fig. 2d and a side-by-side comparison can be found in Supplemental Figure 3.

Minor comments

1. Mixed uses of "Figure" and "Fig."

From the journal rules, we have found that when in the text we should use "Figure", but in brackets we should use "Fig." We have changed the manuscript so that this is done consistently.

2. Mixed uses of "Supplementary" and "Sup"

Thank you. Corrected to "supplementary".

3. Line 110. Typo "Figure 2.a"

Thank you. Corrected.

4. Line 142. Typo "Fig. 3.a"

Thank you. Corrected.

5. claim in 35-37 this could use a citation

This claim has been removed during the process of reworking the introductory section.

8. line 136-137 how do you know it is deaf? Similarly, line 20 how do you know how the stimulus is perceived?

When the ears are popped, we see a collapse of the otic capsule and a displacement of the otoliths from their hair cells. We view this as necessarily disrupting auditory transmission. We now give more details on this manipulation and acknowledge that the deafness has not been experimentally

demonstrated in Lines 152-153.

Reviewer #3 (Remarks to the Author):

Favre-Bulle et al. describe an interesting experimental approach to stimulate the auditory system of zebrafish larvae with optical forces. They use their technique to characterize associated neural activity and the role of the utricular and saccular otoliths.

1. In the abstract the authors say that they use “optical forces to generate localized sound in vivo”, and I was trying to imagine how they would do so in water, in the vicinity of the fish. As it turns out, the authors use an optical trap to vibrate the otoliths of larval zebrafish, allowing controlled stimulation of the auditory organ. In the following the authors compare this stimulation to “true sound” from a speaker. I think it is misleading to distinguish between different kinds of sound in this case. Direct stimulation by vibrating the otoliths is not localized generation of sound. This needs to be clarified.

We have reworked the claims throughout the introduction to make the precise nature of our manipulation clear. Examples can be found in lines 62-67, and 73-79.

2. I am also missing a description of how the oscillation and displacement of the otoliths are induced. There seem to be two alternating optical traps at a given distance. How is this distance determined and what influence does this have on the performance of the oscillator? It is not intuitively clear what this optimal distance should be. Since two shutters are used I assume that the waveform of the frequency is rectangular. Does this result in a sinusoidal oscillation? Is the exact waveform of the oscillation relevant in trying to simulate stimulation by sound waves?

The physical fundamentals of the optical trapping have been laid out in past papers (Favre-Bulle et al, 2015; Favre-Bulle et al, 2018), and we have explained this more clearly in the Methods section (Lines 237-249).

We are not worried of the performance of our mechanical oscillator (galvo mirrors) for these speeds and distances. We have tested its performances ahead of the experiments (in the calibration step) and found that the galvo mirrors performed well for these speeds and distances. Since the dual traps are gated by galvo mirrors, the waveform of the OT is, indeed, rectangular. Due to a variety of physical effects that are difficult to disentangle, including the stiffness of hair cells, inertia of the otoliths, and water viscosity at high frequencies (see first page of supplemental material), the movements of the otoliths are sinusoidal for these frequencies, as would occur during natural sound stimulation.

3. “These vibrations of the otoliths relative to the fish body are similar to what a sound wave would produce”. I cannot find a control experiment that demonstrates that indeed the vibrations induced by the alternating optical trap are similar to what is observed when the ear is hit by a sound wave. How is the frequency response under these natural conditions (relative displacement vs. frequency)? Which direction do the otoliths move relative to the fish? Is this the same direction as you induce with the optical trap?

We have previously performed experiments recording otolith motion to speaker tones, however, as the fish body was also oscillating in response to the mechanical wave, the displacements of otoliths relative to the body was too difficult to measure accurately (See Reviewer 2's Comment 6). We now address the axes of the otoliths' movements in response to Reviewer 2's Comment 3.

4. The figures are of low quality, the graphs are hard to read, and the drawings in Fig. 1 and 2 are very qualitative and not precise enough. Where are the red beams pointing in 2b? Is this to scale? This is not how focused laser beams move when scanned in the BFP. I would also like to see a more detailed optical layout. The 3D rendering is not helpful and does not show the ray paths.

We have reworked Figures 2 and 3 to make them clearer and more visually appealing. We have added a new Supplemental Figure 1 providing details of the optical layout. A detailed description is also found in the Methods (lines 225-244).

5. Overall, the manuscript presents an interesting idea and a nice implementation, however, I am missing a lot of experimental details and measurements that convince me that the stimulation by the optical traps is similar to the response to real sound waves. If this is meant to be a methods paper, more emphasis needs to be put on the justification and characterization of the experimental setup and procedures.

Reviewers' Comments:

Reviewer #2:

Remarks to the Author:

In this study, Favre-Bulle et al. demonstrate an innovative approach to simultaneously stimulate individual otolithic organ using optical forces and measure neural activity. We continue to see lots of potential in using this bio-optic acoustics (BOA) technique to understand mechanisms of the auditory and vestibular pathways, and are excited by this revision.

In the revised manuscript, the authors have addressed the majority of our concerns. In terms of the comparison between effects of acoustic and BOA stimulation, we understand that the authors encountered technical difficulties in measuring the displacement of otoliths under sound stimulation. However, because this is the first time the field has seen BOA, the manuscript needs to be clear about how the authors see BOA stimulation as similar to/different from acoustic stimulation. In light of the limitations on directly comparing the two that the authors detail, this statement (while potentially correct) has insufficient support: "...vibrations of the otoliths relative to the fish body are similar to what a sound wave would produce. [93-94]."

In addition, we suggest the authors add more technical details of the acoustic stimulation performed in Figure 3 (also see comment #6 to the original manuscript). What is the approximate sound pressure level?

We would recommend publication in Nature Communications once the above concerns are addressed and do not need to see the revised manuscript.

Yunlu Zhu, Ph.D. & David Schoppik, Ph.D.

Reviewer #3:

Remarks to the Author:

The authors have addressed all of my concerns. I would only like to see a revision of the abstract to address my original comment no. 1 (the wording "optical forces to generate localized sound in vivo" is misleading and has not been changed) and my comment no. 2. The authors say in their reply "the movements of the otoliths are sinusoidal for these frequencies, as would occur during natural sound stimulation." It would be great if this could be included in the manuscript.

Reviewer #2 (Remarks to the Author):

In this study, Favre-Bulle et al. demonstrate an innovative approach to simultaneously stimulate individual otolithic organ using optical forces and measure neural activity. We continue to see lots of potential in using this bio-optic acoustics (BOA) technique to understand mechanisms of the auditory and vestibular pathways, and are excited by this revision.

1. In the revised manuscript, the authors have addressed the majority of our concerns. In terms of the comparison between effects of acoustic and BOA stimulation, we understand that the authors encountered technical difficulties in measuring the displacement of otoliths under sound stimulation. However, because this is the first time the field has seen BOA, the manuscript needs to be clear about how the authors see BOA stimulation as similar to/different from acoustic stimulation. In light of the limitations on directly comparing the two that the authors detail, this statement (while potentially correct) has insufficient support: "...vibrations of the otoliths relative to the fish body are similar to what a sound wave would produce. [93-94]."

We have shown that OT can vibrate an otolith relative to the body. It is also a known fact that sound vibrations vibrate high density structures (such as otoliths) relative to low density structure (such as the fish body). However, it is true that the equivalence suggested in lines 93-94 (now 98-100) can be confusing for the reader. We have therefore changed it to : "These vibrations of the otoliths within the fish body are also observed with sound waves (Fig. 2a,b) due to the high density of otoliths".

2. In addition, we suggest the authors add more technical details of the acoustic stimulation performed in Figure 3 (also see comment #6 to the original manuscript). What is the approximate sound pressure level?

We have added a section in the Method section about the acoustic stimulation done in Figure 3 (lines 259-268).

We would recommend publication in Nature Communications once the above concerns are addressed and do not need to see the revised manuscript.

Reviewer #3 (Remarks to the Author):

1. The authors have addressed all of my concerns. I would only like to see a revision of the abstract to address my original comment no. 1 (the wording "optical forces to generate localized sound in vivo" is misleading and has not been changed)

We have replaced "sound" by "vibration" to clarify this confusion (line 18).

2. and my comment no. 2. The authors say in their reply "the movements of the otoliths are sinusoidal for these frequencies, as would occur during natural sound stimulation." It would be great if this could be included in the manuscript.

We have added this information to lines 120 to 122.